# Enhanced Recovery and Detection of Highly Infectious Animal Disease Viruses by Virus Capture Using Nanotrap^®^ Microbiome A Particles

**DOI:** 10.3390/v16111657

**Published:** 2024-10-23

**Authors:** Amaresh Das, Joseph Gutkoska, Yadata Tadassa, Wei Jia

**Affiliations:** Reagents and Vaccine Services Section, Foreign Animal Disease Diagnostic Laboratory, National Veterinary Services Laboratories, Animal and Plant Health Inspection Service, United States Department of Agriculture, Plum Island Animal Disease Center, Orient Point, NY 11957, USA; jgutkoska@atcc.org (J.G.); yadata.tadassa@usda.gov (Y.T.); wei.jia@usda.gov (W.J.)

**Keywords:** goatpox virus, sheeppox virus, lumpy skin disease virus, peste des petits ruminants virus, African swine fever virus, virus capture, Nanotrap particles, virus isolation, qPCR/RT-qPCR

## Abstract

This study reports the use of Nanotrap^®^ Microbiome A Particles (NMAPs) to capture and concentrate viruses from diluted suspensions to improve their recovery and sensitivity to detection by real-time PCR/RT-PCR (qPCR/RT-qPCR). Five highly infectious animal disease viruses including goatpox virus (GTPV), sheeppox virus (SPPV), lumpy skin disease virus (LSDV), peste des petits ruminants virus (PPRV), and African swine fever virus (ASFV) were used in this study. After capture, the viruses remained viable and recoverable by virus isolation (VI) using susceptible cell lines. To assess efficacy of recovery, the viruses were serially diluted in phosphate-buffered saline (PBS) or Eagle’s Minimum Essential Medium (EMEM) and then subjected to virus capture using NMAPs. The NMAPs and the captured viruses were clarified on a magnetic stand, reconstituted in PBS or EMEM, and analyzed separately by VI and virus-specific qPCR/RT-qPCR. The PCR results showed up to a 100-fold increase in the sensitivity of detection of the viruses following virus capture compared to the untreated viruses from the same dilutions. Experimental and clinical samples were subjected to virus capture using NMAPs and analyzed by PCR to determine diagnostic sensitivity (DSe) that was comparable (100%) to that determined using untreated (-NMAPs) samples. NMAPs were also used to capture spiked viruses from EDTA whole blood (EWB). Virus capture from EWB was partially blocked, most likely by hemoglobin (HMB), which also binds NMAPs and outcompetes the viruses. The effect of HMB could be removed by either dilution (in PBS) or using HemogloBind™ (Biotech Support Group; Monmouth Junction, NJ, USA), which specifically binds and precipitates HMB. Enhanced recovery and detection of viruses using NMAPs can be applicable to other highly pathogenic animal viruses of agricultural importance.

## 1. Introduction

Rapid and accurate diagnosis of highly infectious viral diseases is key to controlling disease outbreaks in susceptible animals. Devastating infectious animal diseases have huge negative impacts on agriculture and the economy in affected countries. As global trade and transportation become an integral part of the economy and growth of countries, they also risk accidental or intentional (bioterrorism) introduction of diseases into areas with naive livestock that can lead to disease outbreaks. Timely and accurate diagnosis of transboundary animal diseases (TADs) is an important first step to detect, then prevent, control, and stop the spread of diseases. Peste des petits ruminants (PPR), goatpox (GP), sheeppox (SP), lumpy skin disease (LSD), and African swine fever (ASF) are TADs, caused by peste des petits ruminants virus (PPRV), goatpox virus (GTPV), sheeppox virus (SPPV), lumpy skin disease virus (LSDV), and African swine fever virus (ASFV), respectively. The diseases caused by these agents are reportable to the World Organization for Animal Health (WOAH, previously OIE).

PPR is a disease of small ruminants, such as goats and sheep, that is currently considered one of the major TADs with outbreaks in many parts of the world, including Europe, Asia, and Africa [1,2,3,4]. PPRV belongs to the genus *Morbillivirus* in the family *Paramyxoviridae*. At the genomic level, PPRV contains a single-stranded, non-segmented, negative-sense RNA genome of approximately 16 kb [1].

GP, SP, and LSD are the diseases of goats, sheep, and cattle, respectively, caused by GTPV, SPPV, and LSDV, respectively. These viruses belong to the genus *Capripoxvirus* (CaPV) in the family *Poxviridae*. Capripoxviruses (CaPVs) are endemic to many countries in the Asia subcontinent and Africa [5] with recent outbreaks (LSD) reported in many Asian countries [6]. All CaPVs have a double-stranded DNA genome of approximately 150 kb and share 147 putative genes that are highly conserved (96–97%) between the species [7].

ASF is a highly infectious and lethal hemorrhagic viral disease of domestic swine and related wild reservoir hosts, including wild boar, warthogs, and feral swine. The disease is currently present in many countries in Asia and Europe and, more recently, in the Caribbean [8,9]. The causative agent, ASFV, is a DNA virus belonging to the genus *Asfivirus* and the family *Asfarviridae* [10]. The ASFV genome consists of double-stranded DNA of 170–193 kb that contains between 151 and 167 open reading frames (ORFs), depending on the virus strain [10].

Whether routine surveillance or disease outbreak, rapid and sensitive detection of the causative agent(s) is an important first step toward the prevention and control of animal diseases. Molecular detection of viral DNA/RNA by PCR or loop-mediated isothermal amplification (LAMP) is rapid, specific, and sensitive; and the turnaround time is much shorter (~1 to 2 h) than traditional methods such as VI and antigen ELISA. Genomic identification based on the amplification of viral DNA/RNA by PCR or LAMP has been routinely used for rapid detection of PPRV [11,12], CaPVs [13,14,15], and ASFV [16,17,18].

Despite their high sensitivity and specificity, the performance of PCR or LAMP can be compromised by multiple factors, such as PCR inhibition caused by naturally occurring PCR inhibitors or failure to detect the causative agents (virus) in specimens during the initial (asymptomatic) phase of infection when the viral load remains low. In both cases, a true-positive sample can be tested as false-negative, which may lead to the failure of disease control and prevention efforts.

Nanotrap^®^ Microbiome A Particles (NMAPs) are highly porous, thermostable hydrogel particles coupled with chemical affinity baits that can capture and concentrate a broad range of analytes including virions [19,20]. Recently, we have shown that NMAPs can capture and concentrate PPRV from diluted suspensions, and the captured viruses remained infectious and recoverable by virus isolation (VI) using susceptible cell lines (21). In this study, NMAPs were used to capture, concentrate, and recover several highly infectious animal disease viruses, including GTPV, SPPV, LSDV, PPRV, and ASFV. The virus capture and recovery were analyzed by VI and virus-specific qPCR/RT-qPCR.

## 2. Materials and Methods

### 2.1. Virus Strains, Cell Culture, and Virus Isolation

All viruses used in this study were obtained from the biorepository of the Reagents and Vaccine Services Section (RVSS) of the Foreign Animal Disease Diagnostic Laboratory (FADDL) at the Plum Island Animal Disease Center (PIADC). The viruses include GTPV strain Pendik (GTPV-Pendik), SPPV strain HELD (SPPV-HELD), LSDV strain Cameroon (LSDV-Cameroon), PPRV strain Egypt (PPRV-Egypt), PPRV strain Turkey (PPRV-Türkiye), and a Vero-adapted ASFV Lisbon-60 vaccine strain BA71V. A virulent ASFV strain Georgia (ASFV-Georgia) was also used and was provided by the Proficiency and Validation Services Section (PVSS) of FADDL. The TCID_50_/mL titers of the original supernatants of the virus cultures used in virus capture experiments were GTPV, SPPV, LSDV, and PPRV (Egypt) at 10^6.6^, 10^6.6^, 10^6.8^, and 10^5.5^, respectively. The titers of the ASFV strains were not available.

For VI, the viruses were inoculated onto susceptible cells grown on 6-well microtiter plates. Vero E6 cells were used for PPRV and ASFV (BA17V), while primary LK cells were used for GTPV, SPPV, and LSDV. The Vero and LK cells were grown in Eagle’s Minimum Essential Medium (EMEM) supplemented with FBS (10% *v*/*v* for LK and 7% *v*/*v* for Vero) plus an antibiotic/antimycotic cocktail (100×; Gibco/Thermo Fisher Scientific, Waltham, MA, USA) containing penicillin, streptomycin, and amphotericin B at a final concentration (per ml) of 100 units, 100 μg, and 0.25 μg, respectively. The plates were incubated in a CO_2_ incubator (5% *v*/*v*) at 37 °C until the cell density reached 70–75% confluency (2–3 days post-incubation). Freshly prepared cells at the desired confluency (70–75%) were inoculated with the virus for VI. After inoculation, the plates were incubated in a CO_2_ incubator for 1 h for adsorption. Next, the plates were overlaid with EMEM supplemented with 4% FBS plus antibiotics (above) and further incubated in a CO_2_ incubator for an extended period, and the cells were examined for cytopathic effect (CPE) by microscopy (Evos XL Core; Invitrogen/Thermo Fisher Scientific, Waltham, MA, USA).

### 2.2. Animal Experiments and Collection of Diagnostic Samples

Diagnostic samples used in this study were collected either from in-house (PIADC) animal experiments (SPPV, PPRV, or ASFV) or from natural infections (ASFV). Animal experiments were carried out in BSL-3 Ag isolation rooms at PIADC. For SP and PPR, sheep and goats 6–8-month-old, weight 40–60 Ibs, mixed breed, were used, respectively. For ASF, female swine of approximately 60–90 Ibs, Yorkshire breed, were used. Sheep (n = 6) were inoculated intravenously with SPPV-HELD while goats (n = 8) were inoculated intranasally with PPRV-Türkiye. Swine (n = 20) were inoculated intramuscularly (IM) with ASFV-Georgia. All animal procedures were performed following Protocol 225-10-R approved by the Plum Island Animal Disease Center Institutional Animal Care and Use Committee (IACUC), which ensured ethical and humane treatment of experimental animals. After collection of diagnostic samples, the animals (sheep, goats, and swine) were given xylazine intramuscularly to sedate and then Fatal Plus IV to euthanize.

Experimental samples used in the study include swabs (nasal, oral, and conjunctival) immersed in 1 mL of EMEM plus 5% antibiotic/antimycotic in cryovials and EDTA whole blood (EWB) in EDTA tube (Becton, Dickinson and co., Franklin Lakes, NJ, USA). All samples were collected between 7 and 10 days post-inoculation when the animals exhibited mild to severe clinical signs typical for the inoculating viruses including fever, conjunctivitis, and swelling of skin (papules) for sheep (SPPV); fever, diarrhea, and nasal discharges for goats (PPRV); and fever, anorexia, depression, diarrhea staggering gait, and purple skin discoloration for swine (ASFV). The ASFV samples (EWB) were kindly provided by Agriculture Research Services of USDA (USDA-ARS) at PIADC. After collection, all samples were transferred to the laboratory (BSL3) and stored at −70 °C until they were analyzed. The ASFV samples (clinical specimens) were also obtained from the recent ASF outbreaks in the Dominican Republic (DR), which included EWB from naturally infected swine (n = 20) collected in 2023 by the Laboratorio Veterinario Central (LAVECEN) in the Dominican Republic (DR) and kindly provided to us by PVSS (Proficiency and Validation Services) at FADDL. The EWB samples were hemolyzed by freeze–thaw (freeze at −70 °C for 15 min followed by thawing at RT for 15 min) to prevent blood clots prior to use in downstream applications, including virus capture and nucleic acid extractions.

### 2.3. Optimization of Virus Capture Using NAMPs

Previously, we showed that 100 μL of NMAPs (Microbiome A Particles; SKU 44202; CERES Nanosciences Inc., Manassas, VA, USA) was optimum to capture PPRV from diluted suspensions [21]. We repeated the virus capture protocol on other viruses used in this study. A fixed amount of each virus (GTPV, SPPV, LSDV, or ASFV) was diluted in PBS or EMEM in multiple volumes (2-, 5-, 10-, 20-, 50-mL) and then subjected to virus capture using different amounts of NMAPs (50-, 75-, 100-, 150- and 200-μL). The NMAPs and the captured viruses were clarified on a magnetic stand (DynaMag-2, -5, -15, or -50; Thermo Fisher Scientific), reconstituted in PBS (200 μL), extracted (viral DNA/RNA), and analyzed by virus-specific qPCR/RT-qPCR. Based on the Ct values, it was found that the optimum recovery (capture) of the viruses was achieved using 100-, 150-, or 200-μL of NMAPs. Therefore, unless or otherwise stated, 100 μL of NMAPs were used in all virus capture experiments reported onwards.

### 2.4. Assessment of Virus Capture by PCR and Virus Titration

This experiment was carried out using PPRV as the target. Briefly, 200 μL of PPRV-Egypt (original stock; TCID_50_/mL 10^5.5^) was used as the starting material; the virus was diluted 1:50 in EMEM (final volume 10 mL) and then subjected to virus capture using 100 μL of NMAPs. The NMAPs and the captured viruses were clarified on a magnetic stand, washed (1 mL of EMEM), and reconstituted in 200 μL of EMEM. Reconstituted NMAPs and the captured viruses, referred to as “treated” afterward, were serially diluted in EMEM and analyzed separately by PPRV RT-qPCR and virus titration (described below). For positive control (PC), referred to as “untreated” afterward, 200 μL of PPRV (TCID_50_/mL 10^5.5^) was serially diluted in EMEM and analyzed separately by RT-qPCR and virus titration.

Virus (PPRV) titrations were performed on Vero E6 cells grown in 96-well plates. The plates were inoculated with 100 μL of Vero E6 at 10^4^ cells/well and then incubated at 37 °C in a CO_2_ incubator. Once the cells reached 70–75% confluency (~2 to 3 days post-incubation), they were inoculated with serially diluted PPRV (untreated) or reconstituted NMAPs, and the captured viruses (treated) at a rate of 50 μL/well in duplicate. After absorption of the viruses (1 h incubation at 37 °C in a CO_2_ incubator), the plates were overlaid with 50 μL of EMEM (4% FBS plus antibiotics) and re-incubated in a CO_2_ incubator for an extended period (up to 10 days). Monolayers were examined by microscopy (Evos XL Core) for the development of CPE. The virus titers were expressed as Log_10_ TCID_50_/mL based on a calculation described by Cottral [22].

### 2.5. Virus Capture from Suspensions Containing Multiple Viruses

In this experiment, we examined whether NMAPs can capture and concentrate multiple viruses from suspensions. Five viruses used in the study were mixed in three different combinations in PBS in a final volume of 5 mL as follows: GTPV + PPRV + ASFV; SPPV + PPRV + ASFV; and LSDV + PPRV + ASFV. Each virus alone (GTPV, SPPV, LSDV, PPRV, or ASFV) was also diluted in PBS (5 mL final volume) and used as a positive control (PC). The virus suspensions containing either a single virus (homogeneous) or multiple viruses (heterogeneous) were then subjected to virus capture using 100 mL of NMAPs. The NMAPs and the captured viruses were clarified on a magnetic stand, reconstituted (200 μL of PBS), extracted (viral DNA/RNA), and analyzed by virus-specific qPCR/RT-qPCR (Section 2.8 below).

### 2.6. Virus Capture from Whole Blood

In preliminary studies, we found that the viruses captured from whole blood (EWB) were partially blocked, which was most likely due to hemoglobin (HMB) that also bound to NMAPs and outcompeted the viruses. To improve virus capture from EWB (spiked), two different protocols were tested. In one protocol, EWB (spiked with virus) was diluted 1:10 in PBS (pH 7.2; filtered through a 0.2 μ membrane filter) and then subjected to virus capture using NMAPs. In the other protocol, EWB (spiked) was treated with HemogloBind™ (HGB; Biotech Support Group, Monmouth Junction, NJ, USA), which specifically binds and precipitates HMB without interfering with virus capture. EMEM could not be used as a diluent since it caused the formation of blood clots that significantly inhibited virus capture and recovery. EWB from healthy goats, sheep, cattle, or swine (Innovative Research Inc.; Novi, MI, USA) was used in this experiment.

In the dilution protocol, 2 mL of EWB samples (undiluted or diluted 1:10 in PBS) was spiked with the virus (20 μL; original stock) and then subjected to virus capture using 100 μL of NMAPs. For positive control (PC), the same amount of virus (20 μL) was spiked into PBS (2 mL) and then subjected to virus capture using NMAPs. The NMAPs and the captured viruses were clarified on a magnetic stand, washed (1 mL PBS), reconstituted (200 μL PBS), extracted (viral DNA/RNA), and analyzed separately by VI and virus-specific qPCR/RT-qPCR.

In the HGB protocol, EWB spiked with the virus was treated with HGB to remove/precipitate HMB according to the manufacturer’s instructions. Accordingly, EWB from goat, sheep, cattle, and swine was spiked with GTPV or PPRV, SPPV, LSDV, and AFV, respectively. Briefly, 200 μL of EWB was mixed with the virus (20 μL) in a 1.5 mL Eppendorf tube and incubated at RT on a shaker for 5 min for equilibration. Next, 200 μL of HGB was added, and the contents were further incubated at RT on a shaker at RT for 10 min to facilitate binding of HMB to HGB. The contents were centrifuged at 10,000× *g* for 4 min at RT. The precipitate (HMB + HGB) was discarded, and the clear supernatant containing the residual viruses was diluted to 2 mL in PBS and then subjected to virus capture using NMAPs (100 μL). The NMAPs and the captured viruses were clarified on a magnetic stand, washed in PBS (1 mL), reconstituted in PBS or EMEM (for VI), and analyzed separately by qPCR/RT-qPCR and VI.

### 2.7. Virus Capture from Experimental and Clinical Samples and Diagnostic Sensitivity

The NMAPs were used to capture and concentrate viruses from experimental and clinical samples to determine diagnostic sensitivity (DSe). Briefly, 200 μL of swabs or 100 μL of EWB was diluted in PBS (2 mL final volume) and then subjected to virus capture using 100 μL of NMAPs. The NMAPs and the captured viruses were clarified on a magnetic stand, reconstituted (200 μL PBS), extracted (viral DNA/RNA), and analyzed by virus-specific qPCR/RT-qPCR, as described below (Section 2.8). For validation/comparison purposes, nucleic acids (viral DNA/RNA) were also extracted from untreated diagnostic samples as well as negative extraction control (NEC; 200 μL of PBS or NMAPs) and analyzed by virus-specific qPCR/RT-qPCR.

### 2.8. Nucleic Acid (DNA/RNA) Extractions and qPCR/RT-qPCR

The QIAmp^®^ Viral RNA Mini Kit (Germantown, MD, USA) was used for the purification of viral RNA (PPRV), and the Cyclone DNA/RNA Purification Kit (DTPM; Fort Payne, AL, USA) was used for the purification of viral DNA (GTPV, SPPV, LSDV, and ASFV). The latter kit (DTPM) was also used for the purification of viral DNA/RNA from EWB samples. All blood samples (EWB) were hemolyzed by freeze–thaw (freezing at −80 °C and thawing at RT; above) prior to extractions. Briefly, 200 μL of sample was used for each extraction, and the extracted nucleic acids (DNA/RNA) were eluted with 100 μL of elution buffer.

Virus-specific qPCR/RT-qPCR assays were carried out on an Applied Biosystems 7500 Fast thermocycler (Thermo Fisher Scientific). The PPRV RT-qPCR was carried out according to Batten et al. [15] using the Path-ID™ Multiplex One-Step RT-PCR Kit (Thermo Fisher Scientific), as described in [21]. The ASFV qPCR was carried out according to Zsak et al. [19] using TaqMan™ Fast Virus 1-Step Mastermix (Thermo Fisher Scientific). The CaPV (GTPV, SPPV, or LSDV) qPCR assays were carried out using Path-ID™ qPCR Mastermix according to Das et al. [13,14]. The oligonucleotide primers (forward and reverse) for the PCR assays were purchased from Integrated DNA Technology (Coralville, IA, USA), and the TaqMan probes (labeled with FAM as reporter dye at the 5′-end and MGB as the quencher dye at the 3′-end) were purchased from Thermo Fisher Scientific. The PCR reaction master mixes were prepared as per the manufacturer’s instructions to include 1× buffer, enzymes (*Taq* DNA polymerase for qPCR and a reverse transcriptase plus *Taq* DNA polymerase for RT-qPCR), primers, probe, and 5 μL of template (extracted viral DNA/RNA) plus the required amount of nuclease-free water in a final volume of 25 μL. The thermocycling conditions for PCR amplification were as follows:ASFV qPCR: one cycle of 95 °C for 20 s followed by 40 cycles of amplification with each cycle consisting of 95 °C for 10 s and 60 °C for 30 s;CaPV qPCR: one cycle of 95 °C for 10 min (enzyme activation/template denaturation) followed by 40 cycles of amplification with each cycle consisting of 95 °C for 15 s and 60 °C for 60 s;PPRV RT-qPCR: one cycle of 45 °C for 10 min (reverse transcription), one cycle of 95 °C for 10 min (enzyme activation/template denaturation), and 40 cycles of amplification with each cycle consisting of 95 °C for 15 s and 60 °C for 60 s.

## 3. Results

### 3.1. Optimization of Nucleic Acids (Viral DNA/RNA) Extractions

Initially, the QIAmp^®^ Viral RNA Mini kit and Qiagen DNeasy Blood and Tissue kit were used for the purification of viral RNA (PPRV) and viral DNA (GTPV, SPPV, LSDV, or ASFV), respectively. However, the DNeasy Blood and Tissue kit was found to be unsuitable for the extractions of viral DNA from NMAPs and the captured viruses (GTPV, SPPV, LSDV, or ASFV) since the results (Ct values) were either inconsistent or not reproducible. Therefore, all DNA extractions were carried out using the Cyclone DNA/RNA Purification Kit (DTPM). The DTPM kit was also used to extract PPRV RNA. The yield (Ct values) and the amplification efficiency (AE) of the PPRV RNA extracted from serial dilutions using DTPM were found to be comparable to that using the QIAmp^®^ Viral RNA Kit. All PCR results (Ct values) reported in this study also included appropriate negative extraction controls (NECs) including Vero/LK cells or NAMPs (in PBS or EMEM), and they all tested negative (undetermined) by PCR.

### 3.2. Assessment of Virus Capture Analyzed by VI, Virus Titration, and Virus-Specific qPCR/RT-qPCR

Analysis of the NMAPs and the captured viruses by VI (Figure 1 and Figure 2) shows efficient recovery (CPE) of all viruses (GTPV, SPPV, LSDV, PPRV, or ASFV) on susceptible cell lines, and the results are comparable to those obtained using untreated viruses (−NMAPs). The recovery of the viruses by VI was further supported by virus-specific qPCR/RT-qPCR performed on viral DNA/RNA extracted from the supernatants of the corresponding VI cultures.

Recovery of the viruses after capture was further analyzed by titration using PPRV-Egypt as a target. PPRV was randomly selected based on the results (VI and PCR) that all five viruses including PPRV were efficiently captured (~100%) and recovered (VI) using NMAPs. NAMPs and the captured PPRV concentrated from diluted suspensions were reconstituted in EMEM, serially diluted (EMEM or PBS), and analyzed separately by virus titration and RT-qPCR. For PC, the same amount of virus (untreated) was similarly diluted in EMEM or PBS and analyzed separately by virus titration and RT-qPCR. The results (Table 1) show comparable titers (TCID_50_/_mL_) corresponding to each dilution of the virus (treated or untreated), indicating efficient recovery of the viruses using NMAPs. The TCID_50_/_mL_ titer of the highest dilution of the virus was 2.4 or 10^−3^ dilution for the untreated virus (PC) and 1.9 or 10^−4^ dilution for the treated virus. The results of RT-qPCR also showed comparable Ct values corresponding to the viruses from each dilution (treated or untreated). The minor differences in the virus titers or the Ct values corresponding to each dilution (treated or untreated) were within the margin of error. The combined results of virus titration and RT-qPCR indicated efficient recovery of the viruses using NMAPs.

### 3.3. Enhanced Recovery and Sensitivity of Detection of the Viruses Using NMAPs

In this experiment, a fixed amount (200 μL) of appropriately diluted virus (Ct values between 31 and 32) was used as working stock WS (see footnotes of Table 2). The WS of each virus (GTPV, SPPV, LSDV, PPRV, or ASFV) was further diluted in PBS at 1:10 (2 mL), 1:25 (5 mL), 1:50 (10 mL), 1:100 (20 mL), 1:250 (50 mL), 1:500 (100 mL), and 1:1000 (200 mL), and then subjected to virus capture using 100 μL of NMAPs with the exception of the dilutions at 1:500 (100 mL) or 1:1000 dilution (200 mL) where 150 μL of NMAPs was used. The NMAPs and the captured viruses were clarified (magnetic stand), reconstituted (200 μL PBS), extracted (viral DNA/RNA), and analyzed by virus-specific qPCR/RT-qPCR. The results (Table 2) show the viruses were detectable up to the 1:1000 dilution when captured and concentrated using NMAPs (treated), while they were detectable only up to the 1:10 dilution if untreated (-NMAPs), indicating a 100-fold increase in the sensitivity of detection following virus capture.

To further investigate whether NMAPs had any interference in the downstream applications such as nucleic acid extractions or PCR, the viruses were serially diluted in PBS and captured using NMAPs. The NMAPs and the captured viruses were then analyzed by virus-specific qPCR/RT-qPCR, and the results were compared against the serial dilutions of the untreated virus. Three different viruses were used in this experiment including PPRV, ASFV, and LSDV. The linear regression standard curves (Ct values vs. serial dilution; Appendix A) of the amplification of viral DNA/RNA extracted from serial dilutions (LSDV, PPRV, or ASFV) show comparable amplification efficiencies (90–110%) and correlation coefficients (R^2^; >0.99) for both treated and untreated viruses, indicating minimal or no interference of the NMAPs on either extraction or amplification. We would anticipate similar results (amplification efficiencies and correlation coefficients) for other viruses including GTPV and SPPV (not tested) as the latter viruses were also efficiently captured (~100%) and recovered using NMAPs as LSDV and they belong to the same genus and family (described above) with 96–97% identities at the genetic level.

### 3.4. Virus Capture and Recovery from EWB

In this experiment, EWB of healthy goats, sheep, cattle, or swine were spiked with GTPV/PPRV, SPPV, LSDV, or ASFV, respectively, and used as undiluted or diluted (1:10 in PBS) prior to virus capture using NMAPs. Analysis of the NMAPs and the captured viruses by qPCR/RT-qPCR (Table 3) shows no significant changes in the Ct values between the viruses captured from diluted EWB at 1:10 dilution or that that captured from PBS (spiked), while the Ct values were relatively higher for the viruses captured from undiluted EWB. The higher Ct values were most likely due to either PCR inhibition by HMB or suboptimal recovery due to HMB outcompeting the viruses for binding to the NMAPs. Further analysis by VI (Table 4) showed the development of CPE after the first passage (1P) on cells inoculated with the viruses captured from PBS or diluted (1:10) EWBs and after the second passage (2P) on cells inoculated with the viruses captured from undiluted EWBs.

To improve virus capture by chemical treatments, HMB of EWB (spiked) was removed by treatment with HGB followed by centrifugation, and the residual viruses in the supernatants were subjected to virus capture using NMAPs. Analysis of the NMAPs and the captured viruses by VI and qPCR/RT-qPCR (Figure 3; Table 4) showed efficient recovery of the viruses. The NMAPs and the viruses captured from the supernatants of EWB after treatment with HGB developed CPE after 1P (Figure 3 and Table 4), which was the same as for the NMAPs and the viruses captured from diluted (1:10) EWB.

### 3.5. Virus Capture from Experimental and Clinical Samples and Diagnostic Sensitivity

NMAPs were used to capture and concentrate viruses from swabs and EWB of experimentally (SPPV, PPRV, or ASFV) and naturally (ASFV) infected animals and analyzed by PCR to determine diagnostic sensitivity (DSe). The results (Table 5 and Appendix A) show comparable DSe (100%) using either the treated or the untreated samples, indicating efficient recovery and detection of the viruses using NMAPs. The Ct values were found to be slightly higher for the viruses extracted directly from untreated EWB compared to those extracted from treated EWB, which is most likely due to PCR inhibition caused by HMB co-purified with the purified viral DNA/RNA during extractions.

### 3.6. Virus Capture from Suspensions Containing Multiple Viruses

In this experiment, we examined whether NMAPs can capture and concentrate multiple viruses from mixed (heterogeneous) suspensions. All five viruses were used in this study, and they were suspended in PBS in three different combinations: GTPV + PPRV + ASFV, SPPV + PPRV +ASFV, and LSDV + PPRV + ASFV. Viruses were also captured from homogeneous suspensions containing a single virus (GTPV, SPPV, LSDV, PPRV, or ASFV). Analysis of the NMAPs and the captured viruses by PCR (Table 6) showed comparable Ct values of the viruses captured from homogeneous (single virus) and heterogeneous (multiple viruses) suspensions, indicating no change in the efficiency of virus capture from suspensions containing either single or multiple viruses.

## 4. Discussion

One of the objectives of this study was to examine and evaluate the efficacy of Nanotrap^®^ particles to capture and concentrate highly infectious animal disease viruses from diluted suspensions or diagnostic samples to improve their recovery and sensitivity to detection. Nanotrap^®^ particles are known to capture low-abundance targets/analytes from different types of matrices, such as gas, liquids, or biological fluids [19]. NMAPs have been previously used for the enrichment of several infectious human and animal disease viruses, including Rift Valley Fever virus (RVFV), coronaviruses, influenza viruses, and respiratory syncytial viruses [20,23,24,25]. In this study, hydrogel Nanotrap™ particles, also referred to as Nanotrap^®^ microbiome A particles or NMAPs, were used to capture and concentrate several highly infectious transboundary animal disease viruses, including GTPV, SPPV, LSDV, PPRV, and ASFV. Initial optimization of virus capture was carried out using virus suspensions in PBS or EMEM, and the efficiency of virus capture was assessed using virus-specific qPCR/RT-qPCR and VI. The optimized protocol was subsequently used to capture and concentrate viruses from diagnostic (experimental and clinical) samples to determine DSe.

Optimum recovery of the viruses (GTPV, SPPV, LSDV, PPRV, or ASFV) from diluted suspensions was obtained using 100 μL of NMAPs, also reported earlier by us [21] and others [24,25]. All the viruses (GTPV, SPPV, LSDV, PPRV, and ASFV) captured by NMAPs were recoverable by VI using virus-specific susceptible cell lines (Figure 1 and Figure 2), indicating that the infectivity of the viruses was not compromised by NMAPs.

To improve the recovery and sensitivity of detection of the viruses in diluted suspensions, a fixed amount (20 μL) of appropriately diluted virus (GTPV, SPPV, LSDV, PPRV, or ASFV) was further diluted (in PBS) from 10-fold (200 μL) to 10,000-fold (200 mL) and then subjected to virus capture (except 200 μL) using NMAPs and analyzed by virus-specific qPCR/RT-qPCR. The results (Table 2) show untreated viruses were detectable only up to a 10-fold dilution (2 mL), while they were detectable up to a 1000-fold dilution (200 mL) after being captured using NMAPs (treated), a 100-fold increase in the sensitivity of detection compared to the untreated viruses. Furthermore, the NMAPs and the captured viruses from all dilutions exhibited similar Ct values (between 34 and 35), indicating a very similar efficiency (~100%) of virus capture irrespective of the virus or the dilution. Indeed, improved sensitivity of detection (up to 10-fold) of SARS-CoV-2 using NMAPs has also been reported by others [24].

Further assessment of virus capture was carried out by virus titration using PPRV as the target. The results (Table 1) show comparable titers (TCID_50_/_mL_) corresponding to the serial dilutions of the viruses either treated or untreated, further confirming the efficient recovery of the virus using NMAPs.

The amplification efficiency (AE) of the viral DNA/RNA extracted from NMAPs and the captured viruses (LSDV, PPRV, or ASFV) from serial dilutions was determined and compared against that extracted from the same dilutions of the untreated viruses. The linear regression standard curves (Ct vs. serial dilution; Appendix A) showed that AEs were comparable and within the acceptable range (between 90 and 100%) for the viral DNA/RNA extracted from either treated or untreated viruses. The combined results of VI, virus titration, and qPCR/RT-qPCR indicate there were no adverse effects of NMAPs on the cells (Vero or LK) or the infectivity of the viruses, and there was minimal or no interference of the NMAPs on nucleic acid extraction or amplification.

The virus capture (NMAPs) from EWB (spiked) and their analysis by qPCR/RT-qPCR (Table 3) showed partial recovery of the viruses (detectable at higher Cts) from undiluted EWB compared to that from diluted (1:10) EWB. Further analysis of the NMAPs and the captured viruses by VI showed cells inoculated with the viruses captured from diluted (1:10) EWB developed CPE after the first passage, while that captured from undiluted EWB developed CPE after the second passage (Table 4). The delayed recovery (second passage) of the viruses captured from undiluted EWB was most likely due to the partial blocking of virus capture by HMB since it also binds NMAPs and outcompeted the viruses. This was corroborated by the treatment of the EWB with HGB to remove HMB, which resulted in a faster recovery (first passage) (Table 4; Figure 3).

The NAMPs were also used to capture and concentrate viruses (PPRV, SPPV, or ASFV) from experimental and clinical samples to determine DSe. The DSe of the virus-specific qPCR/RT-qPCR on the viruses determined using NMAPs was shown to be comparable (100%) to that determined using untreated samples (Table 5 and Appendix A). The PCR results (Appendix A) also show Ct values were slightly but consistently lower for the viruses captured from EWB compared to those from untreated EWB. The higher Ct values of the viruses from untreated EWB could be due to PCR inhibition by HMB, which was partially separated and removed in the treated EWB after virus capture.

It should be noted that all viruses (GTPV, SPPV, LSDV, PPRV, and ASFV) were captured/recovered from diluted suspensions at very similar efficiencies (~100%), which could have been due to them all being enveloped viruses with a strong affinity for binding to the NMAPs, as previously reported [20,21,24,25]. NMAPs were also shown to capture and concentrate multiple viruses from diluted suspensions (heterogeneous) at similar efficiencies to those from suspensions (homogeneous) containing a single virus (Table 6). Capture and enrichment of multiple respiratory viruses (influenza virus, respiratory syncytial virus, and coronavirus) from specimens using NMAPs have also been reported by others [20]. These findings show that NMAPs can be used to detect multiple viral pathogens in specimens from animals during a co-infection scenario.

One of the major challenges of diagnostic PCR is the false negatives that can occur either due to PCR inhibition by naturally occurring PCR inhibitors or low levels of virus in the samples. Naturally occurring PCR inhibitors present in body fluids or environmental samples include hemoglobin (blood), bile salts and complex polysaccharides (feces), urea (urine), calcium ions (bone, milk), or environmental polysaccharides and humic acid (soil, water) [26,27]. These inhibitors often co-extract and co-purify with the template (DNA/RNA) during extractions and interfere (i.e., PCR inhibition) in the downstream applications (e.g., amplification). There are methodologies/protocols available to neutralize the effect of PCR inhibition, including (1) use of inhibitor-resistant recombinant *Taq* DNA polymerases [28,29] or native thermostable DNA polymerases [30]; (2) use of enhancers or facilitators such as dimethyl sulfoxide (DMSO), betaine, cattle serum albumin (BSA), or glycerol [29]; (3) modification of nucleic acid extraction protocols by adding extra washing steps [31]; or (4) decreasing the inhibitor concentration by sample dilution [32]. Alternatively, the interference of PCR inhibitors can be neutralized by sample dilution followed by virus capture using NMAPs (this study). Therefore, NMAPs can be used for the enrichment and enhanced detection of the viruses in diluted samples, such as environmental samples, where the virus concentration remains below the limit of detection (LOD) of PCR.

Viruses can be concentrated from diluted suspensions/samples by other methods including membrane filtration, skim milk flocculation, polyethylene glycol (PEG) precipitation, or adsorption–extraction [33,34]. These methods can be expensive (equipment) and/or labor-intensive; therefore, they are not applicable in low-resource settings. One of the prime examples of concentrating viruses/pathogens is the wastewater-based surveillance (WBS) of communicable diseases, which has now become one of the most powerful tools used in monitoring public health since the outbreak of SARS-CoV-2 [35,36,37,38,39]. This study shows NMAPs efficiently captured viruses from a wide range of dilutions (1:10 or 1:1000), and therefore concentrating viruses using NMAPs can be more effective than other methods. NMAPs have been successfully used in WBS of several human pathogens, including SARS-CoV-2, monkeypox, enterovirus, human norovirus, human adenovirus, bocavirus, Epstein Barr virus, influenza A virus, and respiratory syncytial virus B [35,36,37,38,39]. Likewise, WBS can be a powerful tool in disease surveillance to monitor highly infectious animal disease viruses in wastewater/discharge from livestock farms, wet markets, or slaughterhouses that are reservoirs and the sites of amplification of infectious agents [40].

NMAPs have been successfully used to monitor (by PCR) virus loads in biological fluids for the detection of several zoonotic and animal disease viruses, including Rift Valley fever virus, Venezuelan equine encephalitis virus, and influenza viruses [41]. One of the potential advantages of the NMAPs is to counter false negatives (PCR) that occur due to either PCR inhibition or a low level of virus (below the LOD of PCR) in the sample. NMAPs can be used to capture, concentrate, and separate viruses from PCR inhibitors in diluted suspensions to enhance recovery and improve their detection. These applications of NMAPs can aid in disease surveillance and accurate diagnosis of animal disease viruses by diagnostic PCR.

## Figures and Tables

**Figure 1 viruses-16-01657-f001:**
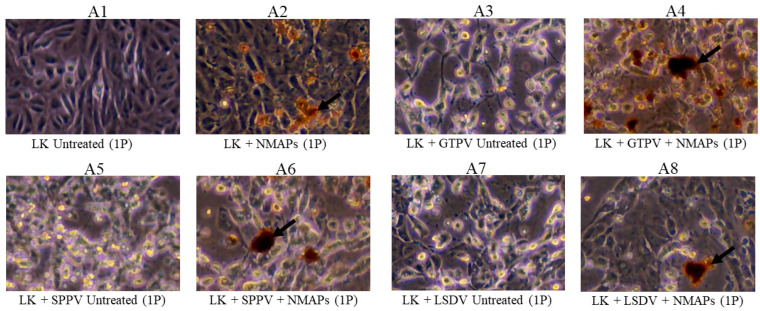
Capture and recovery of GTPV, SPPV, and LSDV from diluted suspensions using NMAPs. Viruses were captured and concentrated using NMAPs as described in the Materials and Methods. The NMAPs and the captured viruses were washed, reconstituted in EMEM, and inoculated onto LK cells for VI and examined by microscopy for the development of CPE for 144 h (1P): (**A3**) cells inoculated with untreated GTPV; (**A4**) cells inoculated with treated (+NMAPs) GTPV; (**A5**) cells inoculated with untreated SPPV; (**A6**) cells inoculated with treated (+NMAPs) SPPV; (A7) cells inoculated with untreated (−NMAPs) LSDV; and (A8) cells inoculated with treated (+NMAPs) LSDV. For negative controls, cells were inoculated with either EMEM (**A1**) or NMAPs (**A2**). NMAPs are shown by arrows. Images were taken under microscope at 20× magnification.

**Figure 2 viruses-16-01657-f002:**
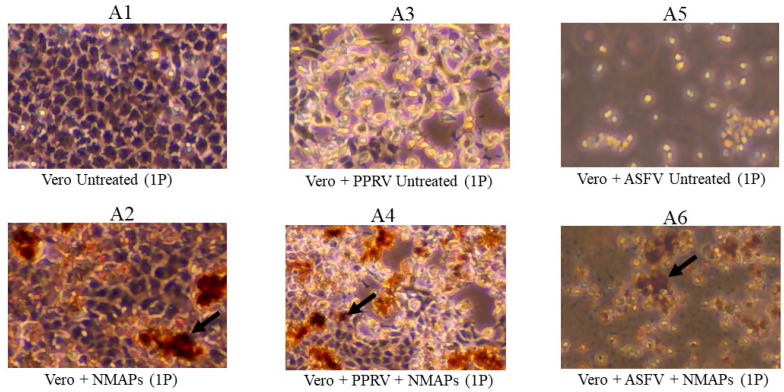
Capture and recovery of PPRV and ASFV from diluted suspensions using NMAPs. Viruses were captured and concentrated using NMAPs as described in the Materials and Methods. The NMAPs and the captured viruses were washed, reconstituted in EMEM, and inoculated onto Vero cells for VI and monitored for the development of CPE by microscopy for up to 168-h (1P): (**A3**) cells inoculated with untreated PPRV; (**A4**) cells inoculated with treated (+NMAPs) PPRV; (**A5**) cells inoculated with untreated ASFV; (**A6**) cells inoculated with treated (+NMAPs) ASFV. For negative controls, cells were inoculated with either EMEM (**A1**) or NMAPs (**A2**). NMAPs are shown by arrows. Images were taken under microscope at 20× magnification.

**Figure 3 viruses-16-01657-f003:**
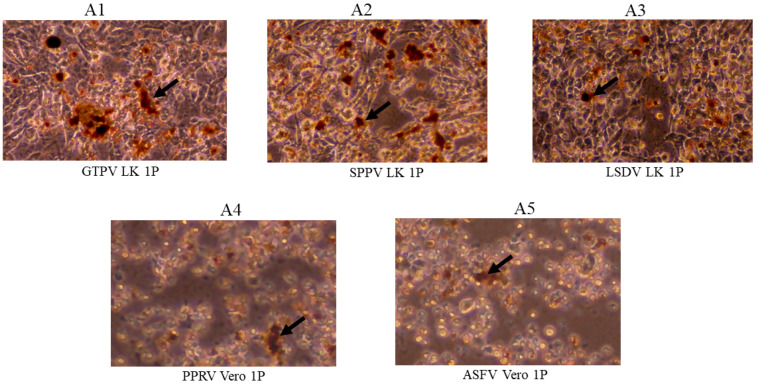
Capture and recovery of residual viruses (spiked) from the supernatants of EWB after treatment with HemogloBind™ (HGB) using NMAPs. Viruses were spiked into EWB of the sensitive host animals and then treated with HGB to precipitate HMB as described in the Materials and Methods (Section 2.6). After clarification (centrifugation), the supernatants containing the residual viruses were diluted in PBS (2 mL) and then subjected to virus capture using NMAPs. The NMAPs and the captured viruses were washed, reconstituted in EMEM, and inoculated onto virus-specific cells for VI (LK for GPV, SPV, and LSDV; Vero for PPRV or ASFV) and were monitored for the development of CPE for 144 h (1P): (**A1**) LK cells inoculated with NMAPs and the captured GTPV; (**A2**) LK cells inoculated with NMAPs and the captured SPPV; (**A3**) LK inoculated with NMAPs and the captured LSDV; (**A4**) Vero cells inoculated with NMAPs and the captured PPRV; and (**A5**) Vero cells inoculated with NMAPs and the captured ASFV. NMAPs are shown by arrows. Images were taken under microscope at 20× magnification.

**Table 1 viruses-16-01657-t001:** Assessment of virus (PPRV) capture analyzed by RT-qPCR and virus titration.

PPRV-Egypt		PPRV RT-qPCR (Cycle Threshold)	Virus Titer (TCID_50_/_mL_)
Serial dilution: Untreated *	Undiluted	17.333	5.5
	10^−1^ Dilution	20.374	4.3
	10^−2^ Dilution	22.914	3.2
	10^−3^ Dilution	26.582	2.4
	10^−4^ Dilution	30.440	Below LOD
	10^−5^ Dilution	32.318	Below LOD
	10^−6^–10^−10^ Dilution	Undetectable	Below LOD
Serial dilution: Treated ^#^	Undiluted	18.547	5.1
	10^−1^ Dilution	20.991	4.5
	10^−2^ Dilution	24.846	3.5
	10^−3^ Dilution	27.147	2.4
	10^−4^ Dilution	29.810	1.9
	10^−5^ Dilution	32.772	Below LOD
	10^−6^–10^−10^ Dilution	Undetectable	Below LOD

* Untreated: 200 μL of PPRV-Egypt (original supernatant; TCID_50_/mL 10^5.5^) was serially diluted in EMEM and analyzed separately by virus titration and PPRV RT-qPCR. ^#^ Treated: 200 μL of PPRV-Egypt (original supernatant; TCID_50_/mL 10^5.5^) was diluted to 20 mL in EMEM and then the viruses were captured and concentrated using 100 μL of NMAPs. The NMAPs and the captured viruses were reconstituted in 200 μL EMEM, serially diluted in EMEM and analyzed separately by virus titration and PPRV RT-qPCR.

**Table 2 viruses-16-01657-t002:** Virus capture from serial dilutions using NMAPs analyzed by qPCR/RT-qPCR.

Working Stock/ Dilution	Cycle Threshold (qPCR/RT-qPCR)									
	PPRV		GTPV		SPPV		LSDV		ASFV	
	UT ^#^	Treated ^¶^	UT	Treated	UT	Treated	UT	Treated	UT	Treated
WS * (0:0)	32.279	ND ^§^	31.754	ND	31.141	ND	32.700	ND	32.289	ND
WS 1:10	35.285	35.543	34.281	34.455	34.295	34.298	35.180	35.756	35.958	35.289
WS 1:25	UD ^†^	35.869	UD	34.722	UD	34.208	UD	35.980	UD	35.448
WS 1:50	UD	35.577	UD	34.929	UD	34.627	UD	35.237	UD	35.238
WS 1:100	UD	35.587	UD	35.993	UD	34.124	UD	35.970	UD	35.316
WS1:250	UD	35.554	UD	34.889	UD	34.688	UD	35.605	UD	35.636
WS 1:500	UD	35.931	UD	34.559	UD	35.943	UD	35.642	UD	35.920
WS 1:1000	UD	35.339	UD	34.986	UD	34.995	UD	35.192	UD	35.509

* WS: Working stock (WS) of the viruses was prepared by appropriately diluting the original supernatant of the virus in PBS to obtain Ct values between 31 and 32. ^#^ UT (Untreated): 200 μL of WS of each virus diluted in PBS at 1:10 (2 mL); 1:25 (5 mL); 1:50 (10 mL); 1:100 (20 mL); 1:250 (50 mL); 1:500 (100 mL); and 1:1000 (200 mL) and 200 μL of aliquot from each dilution was extracted (viral DNA/RNA) and analyzed by virus-specific qPCR/RT-qPCR. ^¶^ Treated: Viruses were captured and concentrated from each dilution (2–200 mL) using 100 mL of NMVPs except for 100 and 200 mL where 150 μL of NMAPs were used. NMAPs and the captured viruses were clarified on magnetic stand, reconstituted (200 μL PBS), extracted (viral DNA/RNA), and analyzed by qPCR/RT-qPCR. Viruses were captured from suspension of volume higher than 50 mL in multiple aliquots of 50 mL on a single 50 mL falcon tube: 2 × 50 mL aliquots for 100 mL or 4 × 50 mL aliquots for 200 mL. ^§^ ND, not determined. ^†^ UD, Undetermined/Undetectable.

**Table 3 viruses-16-01657-t003:** Virus capture from EDTA whole blood (spiked) after treatment with HemogloBind™ (HGB) analyzed by qPCR/RT-qPCR.

Virus	Cycle Threshold (Spiked Virus) of qPCR/RT-qPCR With or Without Virus Capture or HGB Treatment							
	PBS		EWB (0:0)		EWB (1:10)		EWB (0:0) + HGB ^¶^ (Supernatant)	
	Untreated *	Treated ^#^	Untreated	Treated	Untreated	Treated	Untreated	Treated
GTPV	16.907	17.928	21.033	21.375	19.235	18.174	21.439	22.177
SPPV	17.956	18.350	20.231	20.669	20.056	19.523	21.617	22.221
LSDV	17.776	18.711	20.623	19.864	19.742	18.995	21.731	22.108
PPRV	18.561	19.552	20.615	20.718	20.145	19.630	21.846	22.604
ASFV	19.915	20.155	22.033	22.298	19.444	20.603	22.525	23.138

* Untreated: 20 μL of virus (original supernatant) was spiked into 200 μL of PBS or EWB from goat (GTPV/PPRV), sheep (SPPV), bovine (LSDV), or swine (ASFV) and the entire volume (220 μL) was extracted (viral DNA/RNA) and analyzed by virus-specific qPCR/RT-qPCR. ^#^ Treated: 20 μL of virus was spiked into 2 mL of PBS (reference control) or 2 mL of EWB (undiluted or diluted 1:10 in PBS) and then subjected to virus capture using 100 μL of NMAPs. The NMVPs and the captured viruses were clarified on magnetic stand, reconstituted (200 μL PBS), extracted (viral DNA/RNA) and analyzed by virus-specific qPCR/RT-qPCR. ^¶^ HGB treatment: 20 μL of virus was spiked into 200 μL of EWB and then treated with 200 μL of HGB to precipitate HMB according to the manufacturer’s instructions as described in Materials and Methods. After clarification (centrifugation), the pallets (HMB + HGB) were discarded and the supernatants containing the residual viruses were diluted to 2 mL in PBS and subjected to virus capture using 100 μL of NMAPs. The NMAPs and the captured viruses were clarified, reconstituted (200 μL PBS), extracted (viral DNA/RNA), and analyzed by virus-specific qPCR/RT-qPCR.

**Table 4 viruses-16-01657-t004:** Virus capture and recovery from spiked EWB using NMAPs after dilution or treatment with HGB analyzed by VI and qPCR/RT-qPCR *.

Virus (Spiked)	EWB (Host)	EWB (Dilution/Treatment)	NMVPs Added (Y/N)	CPE (VI)		Cycle Threshold (qPCR/RT-qPCR)	
				1P	2P	1P	2P
GTPV	Goat	0:0	Y	N	Y	23.342	18.045
		1:10	Y	Y	Y	17.631	16.495
		HGB (Sup)	Y	Y	Y	18.569	17.663
SPPV	Sheep	0:0	Y	N	Y	23.657	18.322
		1:10	Y	Y	Y	17.913	16.724
		HGB (Sup)	Y	Y	Y	17.812	16.583
LSDV	Cattle	0:0	Y	N	Y	23.644	18.509
		1:10	Y	Y	Y	19.766	18.609
		HGB (Sup)	Y	Y	Y	18.578	17.455
PPRV	Goat	0:0	Y	N	Y	25.257	20.311
		1:10	Y	Y	Y	19.984	18.766
		HGB (Sup)	Y	Y	Y	20.572	19.427
ASFV	Swine	0:0	Y	N	Y	23.712	18.116
		1:10	Y	Y	Y	17.913	16.893
		HGB (Sup)	Y	Y	Y	18.579	17.398

* Viruses were spiked into EWB (undiluted or diluted 1:10 in PBS) of susceptible hosts (animals) and treated with HGB as described in the legend to Table 3. The NMAPs and the captured viruses were clarified on a magnetic stand, washed (PBS), reconstituted (200 μL EMEM), and inoculated onto virus-specific susceptible cell lines for virus isolation (VI) up to 2 passages and monitored for the development of CPE by microscopy.

**Table 5 viruses-16-01657-t005:** Diagnostic sensitivity of virus-specific qPCR/RT-qPCR after virus capture using NMAPs from specimens of experimentally or naturally infected animals *.

Virus	Host Animal	#of Animals	#of Specimens	Infection Route	Ct (qPCR/RT-qPCR)				Diagnostic Sensitivity (%)
					Untreated (−NMAPs)		Treated (+NMAPs)		
					#Pos	#Neg	#Pos	#Neg	
PPRV	Goat	8	28	EI ^#^	28	0	28	0	100
SPPV	Sheep	6	24		24	0	24	0	100
ASFV	Swine	20	20		20	0	20	0	100
ASFV	Swine	20	20	NI ^¶^	20	0	20	0	100

* Experimental samples included nasal, oral, and conjunctiva swabs and EWB collected from goats (PPRV) and sheep (SPV) experimentally infected with PPRV and SPPV, respectively. Experimental samples or specimens of ASFV include EWB collected from swine (n = 20) either experimentally (EI) or naturally infected (NI) with the ASFV (field samples; Dominican Republic). For virus capture, swabs (200 μL) or EWB (100 μL) was diluted to 2 mL in PBS and then subjected to virus capture using 100 μL of NMAPs. The NMAPs and the captured viruses were clarified (magnetic stand), reconstituted (200 μL PBS), extracted (viral DNA/RNA), and analyzed by virus-specific qPCR/RT-qPCR. For untreated specimens, 200 μL of swabs or 100 μL of EWB diluted 1:1 in PBS (200 μL final volume) was extracted (viral DNA/RNA) and analyzed by virus-specific qPCR/RT-qPCR. ^#^ EI, experimental infection. ^¶^ NI, natural infection. Further details of the results including specimen types and the Ct values can be found in Appendix A.

**Table 6 viruses-16-01657-t006:** Virus capture from suspensions containing multiple viruses using NMAPs analyzed by qPCR/RT-qPCR *.

Virus	Virus Suspension (PBS) Single (S)/Mixed (M)	Cycle Threshold (Ct) (qPCR/RT-qPCR)
CaPV	GTPV (S)	27.420 (GTPV)
	GTPV + PPRV + ASFV (M)	27.078 (GTPV)
	SPPV (S)	27.182 (SPPV)
	SPPV + PPRV + ASFV (M)	27.470 (SPPV)
	LSDV (S)	29.754 (LSDV)
	LSDV + PPRV + ASFV (M)	29.116 (LSDV)
PPRV	PPRV (S)	29.001 (PPRV)
	PPRV + GTPV + ASFV (M)	29.157 (PPRV)
	PPRV + SPPV + ASFV (M)	29.893 (PPRV)
	PPRV + LSDV + ASFV (M)	29.846 (PPRV)
ASFV	ASFV (S)	31.659 (ASFV)
	ASFV + GTPV + PPRV (M)	31.397 (ASFV)
	ASFV + SPPV + PPRV (M)	31.785 (ASFV)
	ASFV + LSDV + PPRV (M)	31.307 (ASFV)

* Viruses were appropriately diluted (Ct values between 27 and 32) and used as working stocks (WS). The viruses (WS) were then further diluted in PBS (20 mL) either as a single virus (S) or with multiple (M) viruses in different combinations as shown. The virus suspensions (S or M) were then subjected to virus capture using 100 μL of NMAPs. The NMAPs and the captured viruses were clarified, reconstituted, and analyzed by virus-specific qPCR/RT-qPCR, as described in the legend to Table 5.

## Data Availability

All data are included in the manuscript including Appendix A in support of the original data.

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
