# Peer review of "Enhanced Recovery and Detection of Highly Infectious Animal Disease Viruses by Virus Capture Using Nanotrap® Microbiome A Particles"

_viruses, 2024, doi:10.3390/v16111657_

Round 1
Reviewer 1 Report (New Reviewer)
Comments and Suggestions for Authors
Authors in this paper demonstrates that application of NMAPs to capture viruses from specimens can increase PCR test sensitivity by 100-fold by comparing treated and untreated samples. As such, authors claims that this process will be useful in surveillance program when sample may contain low virus load (asymptomatic animal) or in disease diagnosis where PCR inhibitor may lead to false negative.
Experiments are well designed; Results are well presented. A couple of points need to be further illustrated:
1. Antibiotics concentrations in cell culture medium: 20% v/v will make 5% of antibiotics. Antibiotics at this level usually will be toxic to cells. What are the concentrations in terms of units.
2. potential applications: in a naturally occurred disease, samples contain abundant viruses. Virus loads are so high so that PCR inhibitors may not be irrelevant. Most cases animals contracted viruses already shed high level viruses before typical symptoms appear. Only the environmental samples may contain low virus loads, which may need virus enrichment.
Author Response
Responses to reviewer1
Authors in this paper demonstrates that application of NMAPs to capture viruses from specimens can increase PCR test sensitivity by 100-fold by comparing treated and untreated samples. As such, authors claims that this process will be useful in surveillance program when sample may contain low virus load (asymptomatic animal) or in disease diagnosis where PCR inhibitor may lead to false negative.
- We thank the reviewer for the positive assessment of this work which is highly appreciated!
Experiments are well designed; Results are well presented. A couple of points need to be further illustrated:
- Antibiotics concentrations in cell culture medium: 20% v/v will make 5% of antibiotics. Antibiotics at this level usually will be toxic to cells. What are the concentrations in terms of units.
- We thank the reviewer for this important correction on the concentration of antibiotic/antimycotic cocktail used in the medium. The concentration of the antibiotics/antimycotic are now expressed in units or mg per ml in the revised manuscript as follows:
Original text: “The Vero and LK cells were grown in Eagle’s Minimum Essential Medium (EMEM) supplemented with FBS (10% v/v for LK and 7% v/v for Vero) plus 20% (v/v) of anas antibiotic/antimycotic cocktail containing penicillin, streptomycin, and amphotericin B (Gibco/Thermo Fisher Scientific)”
Revised text: “The Vero and LK cells were grown in Eagle’s Minimum Essential Medium (EMEM) supplemented with FBS (10% v/v for LK and 7% v/v for Vero) plus an antibiotic/antimycotic cocktail (100x; Gibco/Thermo Fisher Scientific) containing penicillin, streptomycin, and amphotericin B at a final concentration (per ml) of 100 units, 100 ug, and 0.25 mg, respectively”
- Potential applications: in a naturally occurred disease, samples contain abundant viruses. Virus loads are so high so that PCR inhibitors may not be irrelevant. Most cases animals contracted viruses already shed high level viruses before typical symptoms appear. Only the environmental samples may contain low virus loads, which may need virus enrichment.
- We agree with the reviewer on this comment that concentrating viruses from specimens using nanoparticles might not be a good practical approach in symptomatic animals where the viruses already shed at high level. Therefore, we revised the sentence (Discussion) as follows:
Original text: “Therefore, NMAPs can be used to enhance the detection of viruses in biological fluids containing PCR inhibitors or in diluted samples where the virus concentrations remain below the limit of detection (LOD) of PCR as in the specimens from animals at the early stages of infection”
Revised text: “Therefore, NMAPs can be used for the enrichment and enhanced detection of the viruses in diluted samples, such as, environmental samples, where the virus concentration remains below the limit of detection (LOD) of PCR”

Reviewer 2 Report (New Reviewer)
Comments and Suggestions for Authors
The authors present a manuscript which details the use of a magnetic microbead capture technology for enrichment of viral particles from either dilute pure culture or whole blood samples. The authors find that the beads work quite well from pure culture, and consistently saturate available binding sites with virus. Results from whole blood are somewhat less sensitive, but this is to be expected given the presence of hemoglobin, a known competitor for binding sites on this particular microbead. The manuscript is well laid out and easy to follow. Grammar and syntax are nearly perfect, with no need for any additional editing, as far as I can tell. The methods are well established, and have been followed and utilized correctly. The overall findings are incremental, but never the less useful for the broader virology community. I see no reason not to accept the manuscript in the current form.
Author Response
Responses to reviewer2
- We thank the reviewer for positive comments on our manuscript and the recommendation for publication!
This manuscript is a resubmission of an earlier submission. The following is a list of the peer review reports and author responses from that submission.
Round 1
Reviewer 1 Report
Comments and Suggestions for Authors
Line 51-52: Review and rewrite the latest ICTV classification
Please re-check and re-edit the entire manuscript according to the rules of scientific writing (such as the use of exponentials and subscripts; carbon dioxide, degrees, etc., ) (“The TCID50/mL titers of the original supernatants of the virus cultures used in virus capture experiments were GTPV, SPPV, LSDV, and PPRV (Egypt) at 106.6, 106.6, and 106.8, and 105.5, respectively…”) . In many places I guess that the microlitre sign or division signs are not visible. Therefore, there are difficulties of understanding in many parts of the article.
Line 120 and others: According to the United Nations, Turkey is written as Turkiye.Please correct whole manuscript according to UN.
Please prepare the figures and tables according to the journal rules. It looks like screenshots of tables and figures have been taken and added. It should be prepared again in high quality resolution and according to the journal rules.
Table3: The untreated values for EWB 1:10 are not shown in Table 3, but their inclusion in the table is important for comparison.
Line 556: There is no 44th reference, but the authors write in the manuscript.
There is no information about negative ext control with nmaps. Sharing negative extraction Ct values with NMAPS among studies will help comparison of high Ct values.
Could you prepare a standard curve with 3 replicate samples? If the authors have already run with 3 replicates, please indicate this in the manuscript and share the standard curve figures.
Author Response
Reviewer’s comments and Authors response:
Reviewer 1
My major concern is that the authors don’t quite understand what it takes to carry out laboratory testing for the pathogens in a routine member. The way the introduction is made does necessarily justify the use of the beads. Routine diagnostics involves different matrices, with brain, feces and blood being challenging. But the inhibitory effects are easily overcome by resistant Taq polymerases and magnetics beads when DNA/DNA is extracted.
- We are little surprised to hear about negative comments from the reviewer on how we carry out laboratory testing of pathogens. We assume this negative comment might be directed to not providing details of animal experiments being carried out in this work which is now being provided in section 2.2. of materials and methods in the revised manuscript.
- Yes, inhibitor resistant Taq DNA polymerases (not common in commercial PCR kits) can be used to partially overcome PCR inhibition. However, these polymerases not readily available commercially and they are expensive.
- Here we have shown that the detection of the viruses in whole blood can be improved by dilution followed by virus capture with NAMPs using commercially available PCR kits.
Samples received by a laboratory for such reportable pathogens are rarely in low titer, no need to enrich them.
- This may be true when the samples are received from disease outbreaks when the infected animals are severely sick. However, in cases of disease surveillance or during the early stages of infection (pre-clinical) the virus concentrations remain low, often below the limit of detection of the diagnostic PCR resulting false negatives. The ability of the NMAPs to capture and concentrate viruses from diluted suspensions can be a solution to this problem as shown in Table 2 where the viruses were detectable up to 1:1000 dilution after virus capture (+NMAPs) but not detectable using untreated (-NMAPs) samples.
There are seldom cases when the viral RNA content is inconclusive, again this is overcome using routine techniques and the diagnosis is made. By the way false negative results are controlled by internal positive controls. So, beads fail to come any closer to solve problems in this particular setting.
- False negative results due to PCR inhibition can be confirmed using internal positive control but that does not provide any solution to resolve the problem. The ability of the NAMPs to capture, concentrate and separate viruses from the inhibitors can limit their influence and simultaneously improve the limit of detection of the viruses by PCR.
The authors are very fuzzy about the how the samples from animals were collected, nothing about the excremental design, clinical signs of diseased animals. Animals were used, any ethics, the mode of euthanasia???
- We thank the reviewer for this constructive comment. We extensively revised section 2.2 of the revised manuscript to include details of the animal experiments that being conducted in this study and all experiments were carried out using approved animal protocol mandated by Institutional Animal Care and Use Committee (IACUC).
The definition “clinical specimen” can only be applied to samples from a real field outbreak. Throughout the manuscript those should be experimental samples. Very misleading
- We agree with the reviewer and replace the term “clinical specimens “with “experimental samples” for experimentally infected animals and used the term “clinical specimens” for naturally infected (ASFV) animals (swine) in the revised manuscript (section 2.2).
L523 When it comes to capripoxviruses, ASF, PPR, “feces, urea (urine), calcium ions (milk), or environmental polysaccharides (soil)” don’t make sense at all
- We could not find this typo in the manuscript. We assume, this typo was automatically generated when the text file (word document) was converted into a pdf format, and we are not checking this typoe prior to submission of the manuscript to the journal for review.
Overall, the justification for the study looks a big stretch. The materials and methods are not described properly.
- Since the reviewer did not specify any particular concern, we assume it is related to the earlier comments on the lack of details of animal experiments. We extensively revised section 2.2. of materials and methods to include details of animal experiments addressing reviewer’s concerns in the revised manuscript.
Such beads would work for waste waters and animal feces, where inhibition may be a problem, for routine diagnostics – no.
- We respectfully disagree with this comment of the reviewer since naturally occurring PCR inhibitors are ubiquitous, and present not only in animal feces but also in sample types including blood, tissues, urine, soil, milk etc. NAMPs can work similarly (as in wastewater or animal feces) on other samples as above by separating the viruses from the PCR inhibitors thereby limiting their influence (inhibition) in diagnostic PCR.
Reviewer 2 Report
Comments and Suggestions for Authors
The authors describe the use of nanotrap beads to aid in the laboratory detection of animal pathogens using PCR. Nanotraps have been described previously and this study is a follow up to what was done in the past.
My major concern is that the authors don’t quite understand what it takes to carry out laboratory testing for the pathogens in a routine member. The way the introduction is made does necessarily justify the use of the beads. Routine diagnostics involves different matrices, with brain, feces and blood being challenging. But the inhibitory effects are easily overcome by resistant Taq polymerases and magnetics beads when DNA/DNA is extracted.
Samples received by a laboratory for such reportable pathogens are rarely in low titer, no need to enrich them. There are seldom cases when the viral RNA content is inconclusive, again this is overcome using routine techniques and the diagnosis is made. By the way false negative results are controlled by internal positive controls. So beads fail to come any closer to solve problems in this particular setting.
The authors are very fuzzy about the how the samples from animals were collected, nothing about the excremental design, clinical signs of diseased animals. Animals were used, any ethics, the mode of euthanasia???
The definition “clinical specimen” can only be applied to samples from a real field outbreak. Throughout the manuscript those should be experimental samples. Very misleading
L523 When it comes to capripoxviruses, ASF, PPR, “feces, urea (urine), calcium ions (milk), or environmental polysaccharides (soil)” don’t make sense at all
Overall, the justification for the study looks a big stretch. The materials and methods are not described properly.
Such beads would work for waste waters and animal feces, where inhibition may be a problem, for routine diagnostics – no.
Author Response
Reviewers comments and Authors Response:
Reviewer 2
The manuscript is written correctly and presents relevant results in accordance with the proposed objective. However, there are some points that need to be revised. These points are as follows:
- In line 130: the symbol “µ” is missing; this omission is repeated when mentioning all of volume used (e.g., line 137, 140, 147, and throughout the manuscript). Please review.
- We thank the reviewer finding this typo which was automatically generated when the text file (word) was converted into pdf format and regretfully we overlooked these typos before submission of manuscript to the journal. These typos are being corrected in the revised manuscript.
- Line 142: Was the choice of PPRV as a target of virus capture by RT-qPCR and titration made at random? Please explain why this virus was chosen as the target over others mentioned in the manuscript.
- PPRV was randomly selected based on the results (VI and PCR) that all five viruses including PPRV were very efficiently captured (~100%) and recovered (VI) using NMAPs. Based on these results (PCR and VI) we would assume to have similar results using either of the remaining viruses used in this study.
- In line 154: the degree symbol should be in superscript as “°C”.
- Again, we thank the reviewer finding this typo which is now being corrected in the revised manuscript.
- The results obtained and mentioned in the lines from 227 to 232, Materials and methods section, should be transfer to Results section.
- We transfer these lines to the results section as suggested.
Reviewer 3 Report
Comments and Suggestions for Authors
The manuscript entitled "Enhanced Recovery and Detection of Highly Infectious Animal Disease Viruses by Virus Capture Using Nanotrap® Microbiome A Particles" was evaluated.
The manuscript is written correctly and presents relevant results in accordance with the proposed objective. However, there are some points that need to be revised. These points are as follows:
- In line 130: the symbol “µ” is missing; this omission is repeated when mentioning all of volumen used (e.g., line 137, 140, 147, and throughout the manuscript). Please review.
- Line 142: Was the choice of PPRV as a target of virus capture by RT-qPCR and titration made at random? Please explain why this virus was chosen as the target over others mentioned in the manuscript.
- In line 154: the degree symbol should be in superscript as “°C”.
- The results obtained and mentioned in the lines from 227 to 232, Materials and methods section, should be transfer to Results section.
Author Response
Reviewers comments and Authors response:
Reviewer 3
Line 51-52: Review and rewrite the latest ICTV classification
- We checked the latest ICTV section and make the necessary corrections in the revised manuscript.
Please re-check and re-edit the entire manuscript according to the rules of scientific writing (such as the use of exponentials and subscripts; carbon dioxide, degrees, etc.,)
- We thank the reviewer finding these typos. As mentioned above, these typos were automatically generated during the conversion of the text file (word) into pdf format and regretfully, we failed to correct them before submitting the manuscript to the journal. These typos are being corrected in the revised manuscript.
(“The TCID50/mL titers of the original supernatants of the virus cultures used in virus capture experiments were GTPV, SPPV, LSDV, and PPRV (Egypt) at 106.6, 106.6, and 106.8, and 105.5, respectively…”) . In many places I guess that the microlitre sign or division signs are not visible. Therefore, there are difficulties of understanding in many parts of the article.
- We thank the reviewer again finding these typos that were generated in the pdf file when converted from the text file and regretfully we overlooked those typos before submitting the manuscript to the journal. These corrections are being taken care of in the revised manuscript.
Line 120 and others: According to the United Nations, Turkey is written as Turkiye. Please correct whole manuscript according to UN.
- This correction is being made in the revised manuscript.
Please prepare the figures and tables according to the journal rules. It looks like screenshots of tables and figures have been taken and added. It should be prepared again in high quality resolution and according to the journal rules.
- This anomaly (screen shots) applies to Table 2 and all figures, and they are corrected in the revised manuscript including Table 2 (change it to word and changing the orientation from portrait to landscape) and figures (revised figures are now TIF files).
Table3: The untreated values for EWB 1:10 are not shown in Table 3, but their inclusion in the table is important for comparison.
- The Ct values corresponding to untreated EWB 1:10 has been added in the revised Table 3 as suggested.
Line 556: There is no 44th reference, but the authors write in the manuscript.
- Once again, we thank the reviewer finding this typo. There was a missing number of a reference (#8) referred to WOAH (https://www.woah.org/app/uploads/2024/04/asf-report49.pdf) that automatically reduced the number of references to 43 instead of 44. The corrections are being made in the revised manuscript.
There is no information about negative ext control with nmaps. Sharing negative extraction Ct values with NMAPS among studies will help comparison of high Ct values.
- Yes, we agree with the reviewer, the negative extraction controls (NEC) are as important as the positive controls. We did not include any data corresponding the NEC in the manuscript since they were negative (undetermined) and that applies to NEC using NMAPs in PBS or EMEM or cells (LK or Vero). This information is being added to section 3.1. and in the legends to supplemental figure 1 of the revised manuscript.
Could you prepare a standard curve with 3 replicate samples? If the authors have already run with 3 replicates, please indicate this in the manuscript and share the standard curve figures.
- Yes, we have already prepared standard curves on the viral DNA/RNA extracted from serial dilutions serial dilutions of three viruses (PPRV, LSDV and ASFV in 3 replicates ) that were shown in supplemental figure 1.
Round 2
Reviewer 1 Report
Comments and Suggestions for Authors
The authors didn't respond to my comments, so I waited. The author made changes to the manuscript, but they didn't upload a version that we could track. So in some parts of the manuscript they used yellow, grey, underlined or bold, but they didn't mention some of the changes. The new version was too difficult to check. They didn't give enough time for revision, there were a lot of writing mistakes for a revised manuscript.
In my previous comment I mentioned ICTV. Please write virus names according to ICVT. There is a section about when to write it in italics.
In my previus comment I wrote "Please re-check and re-edit the entire manuscript according to the rules of scientific writing (such as the use of exponentials and subscripts; carbon dioxide, degrees, etc., )". But the authors have changed some of them, so please check the whole manuscript and don't use 0 (zero) for degrees.
Table 2: The author didn't check the manuscript They wrote "aWS, bUT..". Please write it correctly. .
Table 4: Could the authors write/discuss why they couldn't detect viruses on 2nd passage, they observed CPE but no Ct?
Reviewer 2 Report
Comments and Suggestions for Authors
The previous concerns not addressed at all.
